# Enforcing boundary conditions for physics-informed neural operators

## Abstract

Machine-learning based techniques like physics-informed neural networks (PINNs) and physics-informed neural operators (PINO) are becoming increasingly adept at solving even complex systems of partial differential equations (PDEs). Boundary conditions can be enforced either weakly by penalizing deviations in the loss function or strongly by training a solution structure that inherently matches the prescribed values and derivatives. The former approach is easy to implement but the latter can provide benefits with respect to accuracy and training times. However, previous approaches to strongly enforcing Neumann or Robin boundary conditions require a domain with a fully $C^1$ boundary and, as we demonstrate, can lead to instability if those boundary conditions are posed on a segment of the boundary that is piecewise $C^1$ but only $C^0$ globally. We introduce a generalization of the approach by Sukumar, et al. (2021) and a new approach based on orthogonal projections that overcome this limitation. The performance of these new techniques is compared against weakly and semi-weakly enforced boundary conditions for the scalar Darcy flow equation and the stationary Navier-Stokes equations.

## 1 Introduction and related work

Various machine learning-based techniques have been applied successfully to solve systems of partial differential equations (PDEs). Most prominently, physics-informed neural networds Raissi, et al. (2019) and many variants and extensions Raissi, et al. (2024) learn the solution to a PDE. They use (often dense) neural networks combined with loss functions that include the PDE residuals to promote physically meaningful solutions and reduce the amount of training data required. Neural operators Li, et al. (2020a); Lu, et al. (2021a) were developed to learn solution operators of PDEs. Based on the Fourier neural operator (FNO) by Li, et al. (2020b), a physics-informed neural operator (PINO) was proposed by Li, et al. (2021). Similar to PINNs it was designed to approximate the solution operator of a parametric partial differential equation by minimizing a residual given by the differential equation instead of training solely on labeled training data. In the following we utilize the FNO framework even though it requires a rectangular domain with a uniform mesh to use the Fast Fourier Transform (FFT). To work on more complex geometries we follow the approach by Lu, et al. (2021c) and choose the minimum bounding box of the underlying domain as computational domain. Alternatively, for more efficient approaches one could utilize the geo-FNO framework proposed by Li, et al. (2022) or geometry-informed neural operators Li, et al. (2023).

**Boundary conditions.** In physics-informed machine learning, boundary conditions can be enforced in two ways. One is to weakly enforce them by adding a residual term that punishes but does not prohibit differences to the prescribed values. An extension are penalty methods to treat boundary conditions as hard constraints in the optimization Lu, et al. (2021b). However, Toscano, et al. (2025); Zeinhofer, et al, (2024) show that these approaches weaken the decay of the generalization

error. The alternative is to strongly enforce boundary conditions by constructing the solution in a way that it exactly satisfies the boundary conditions. While this is straightforward for Dirichlet boundary conditions Berrone, et al. (2023); Toscano, et al. (2025), Neumann- or Robin boundary conditions are harder to treat. Techniques to do this include Fourier feature embeddings Straub, et al. (2025) or solution structures based on trial solutions using distance functions Manavi, et al. (2024); McFall, et al. (2009). Based on the latter idea, Sukumar, et al. (2021) introduce a flexible method to strongly enforce boundary conditions and observe that it can improve accuracy for PINNs. We will show that their approach is suitable for a certain class of boundary conditions but fails when Neumann conditions are prescribed on a boundary that is not $C^1$. They utilize the theory of R-functions by Rvachev, et al. (1995) and approximate distance functions to train solutions to boundary value problems using PINNs. An approximate distance function satisfies two properties.

**Definition 1.1** (Distance function). Let $\Gamma \subset \partial\Omega$ be a boundary section. We call a function $\phi : \bar{\Omega} \to \mathbb{R}$ a *distance function to* $\Gamma$ if it is zero on $\Gamma$ and positive in $\bar{\Omega} \setminus \Gamma$.

The other definition is according to Rvachev, et al. (1995).

**Definition 1.2** (Normalized function). Let $\Gamma \subset \partial\Omega$ be a boundary section. We call a function $\bar{\phi} : \bar{\Omega} \to \mathbb{R}$ a *normalized function with respect to* $\Gamma$ if it satisfies $\bar{\phi} \equiv 0$ and $\frac{\partial \bar{\phi}}{\partial \nu} \equiv 1$ on $\Gamma$.

Here, $\nu$ denotes the inward pointing normal vector on $\partial\Omega$.

**Definition 1.3** (Approximate distance function). Let $\Gamma \subset \partial\Omega$ be a boundary section. We call a function $\phi : \bar{\Omega} \to \mathbb{R}$ an *approximate distance function to* $\Gamma$, if $\phi$ is both a distance function to $\Gamma$ in the sense of Definition 1.1 and normalized with respect to $\Gamma$ in the sense of Definition 1.2.

As pointed out by Sukumar, et al. (2021), suitable approximate distance functions should be $C^1$. Otherwise, the Laplacian of the distance function is unbounded at points where the distance function is only $C^0$, which causes issue when solving second-order differential equations. While the exact distance function $d(\boldsymbol{x}) := \min_{\tilde{\boldsymbol{x}} \in \partial\Omega} ||\boldsymbol{x} - \tilde{\boldsymbol{x}}||$ is an approximate distance function to $\partial\Omega$, it is in general not $C^1$. Sukumar, et al. (2021) discuss how to construct approximate distance functions to boundaries that consist of piecewise linear segments which are only $C^0$ globally. Their approximate distance functions are $C^1$ in the interior but not at the joining points of the segments. This is not problem for the Dirichlet conditions or Neumann boundary conditions on the annulus they consider, where the boundary is globally $C^1$. However, as pointed out by Gladstone, et al. (2022), the non-differentiability of the approximate distance function becomes an issue for Neumann or Robin boundary value on boundaries that are not globally $C^1$. We provide a summary of Sukumar, et al. (2021)'s approach in Appendix A.

To illustrate the issue that can arise if $\partial\Omega \notin C^1$, consider a simple Poisson problem with homogeneous Neumann boundary condition

$$\forall (x, y) \in (0, 1)^2 : \quad -\Delta u(x, y) = 2\pi^2 \cos(\pi x) \cos(\pi y). \tag{1}$$

Figure 1 shows resulting solution (middle) as well as the analytical solution (left) and the solution using our generalized approach presented in this paper (right) for comparison. The issue in the middle figure stems from the emergence of instabilities in the corner of the Laplacian of the approximate distance function, cf. Sukumar, et al. (2021, Figure 27).

**Contributions and structure of the paper.** We propose two novel approaches to prescribe Robin or Neumann boundary conditions on boundaries that are piecewise $C^1$ but only $C^0$ globally. First we describe a generalization of the method by Sukumar, et al. (2021) called *generalized local solution structures* or GLSS. The second approach is based on *orthogonal projections* and we refer to it as OP. While it requires certain assumptions on the shape of the boundary, it has fewer unknown functions that need to be learned by the network. GLSS and OP are described in § 2.1 for scalar PDEs and in § 2.2 for systems of PDEs. § 3.1 compares GLSS, OP as well as weakly and semi-weakly boundary conditions for a scalar PDE, the Darcy flow equation. Finally, § 3.2 compares their performance for a standard benchmark from computational fluid dynamics by Turek, et al. (1996) that requires solving the stationary Navier-Stokes equations to model flow around a cylinder.

**Limitations.** The two new approaches come with some limitations and drawbacks. First, they increase complexity of implementation compared to weakly enforced boundary conditions, in particular if the number of $C^1$-segments that form the boundary is high. Second, while the size of the network

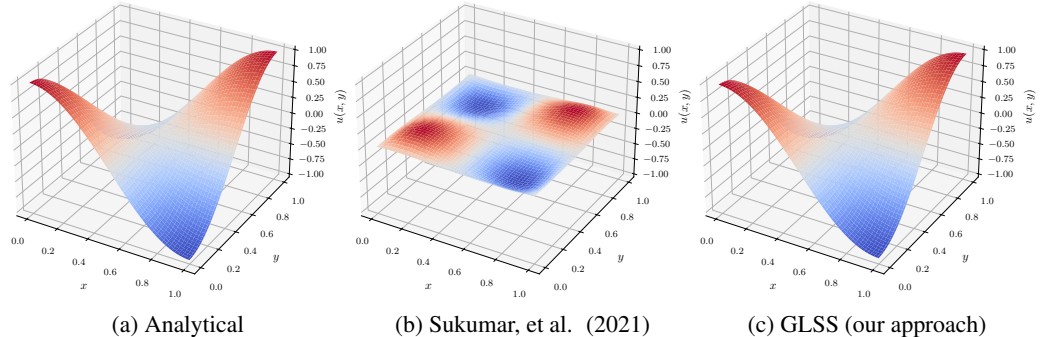

| (a) Analytical | (b) Sukumar, et al. (2021) | (c) GLSS (our approach) |

Figure 1: Analytical solution (left) and solution with Sukumar, et al. (2021)'s (middle) and our generalized approach (right) to strongly enforcing boundary conditions.

as well as training times change only marginally, we do see an increase of inference times of up to 30%. Third, the approaches are only tested for $\Omega \subset \mathbb{R}^2$, although generalization to 3D should be possible. And lastly, the OP approach only works for boundaries that can be decomposed into segments where each of these segments lies in a hyperplane.

## 2   Methodology

Let $\Omega \subset \mathbb{R}^2$ be a computational domain, $\mathcal{U}, \mathcal{V}$ Banach spaces, and let $\mathcal{A} \subset \mathcal{V}$ be a set of parameter functions. For a $\boldsymbol{a} \in \mathcal{A}$ we want to find the solution $\boldsymbol{u} \in \mathcal{U}$ to the boundary value problem

$$\forall \boldsymbol{x} \in \Omega : \qquad\qquad \mathcal{P}(\boldsymbol{u}(\boldsymbol{x}), \boldsymbol{a}(\boldsymbol{x})) = 0, \qquad\qquad (2a)$$

$$\forall \boldsymbol{x} \in \partial\Omega : \qquad\qquad \mathcal{B}(\boldsymbol{u}(\boldsymbol{x}), \boldsymbol{a}(\boldsymbol{x})) = 0, \qquad\qquad (2b)$$

where $\mathcal{P}$ is a differential operator and $\mathcal{B}$ is a boundary condition operator. The idea by Sukumar, et al. (2021) and Rvachev, et al. (1995) is to train suitable functions $\Psi_i$ such that $\tilde{\boldsymbol{u}}(\Psi_1, \ldots, \Psi_I)$ minimizes the PDE-loss from (2a) and, by construction, satisfies the boundary condition exactly.

**Definition 2.1** (Solution structure). We call $\tilde{\boldsymbol{u}} : \{\bar{\Omega} \to \mathbb{R}^I\} \to \{\bar{\Omega} \to \mathbb{R}^n\}$ a solution structure, if $\tilde{\boldsymbol{u}}(\Psi_1, \ldots, \Psi_I)$ satisfies (2b) for any differentiable functions $\Psi_1, \ldots, \Psi_I : \bar{\Omega} \to \mathbb{R}$.

**Physics-informed neural networks (PINNs) and physics-informed neural operators (PINO).** For a PINO, the aim is to learn the solution operator $\mathcal{G}_{\boldsymbol{\theta}} : \mathcal{A} \to \mathcal{U}$ with $\mathcal{G}_{\boldsymbol{\theta}}(\boldsymbol{a}) = \boldsymbol{u}$ using a set of training parameters $\mathcal{A}_{\text{train}} \subset \mathcal{A}$. We denote the trainable parameters of the neural operator as $\boldsymbol{\theta}$. Note that Sukumar, et al. (2021) present their approach in the context of physics-informed neural network (PINNs) whereas we consider physics-informed neural operators (PINO). However, in the notation above, a PINN learns $u_\theta(\boldsymbol{a}^\star) = G_\theta(\boldsymbol{a}^\star)$ for a *fixed* parameter $\boldsymbol{a}^\star$. That is, it learns one specific instance $G_\theta(\boldsymbol{a}^\star)$ solving the boundary problem (2a). By contrast, a PINO trains on a large set of parameters to learn to mapping $\boldsymbol{a} \mapsto G_\theta(\boldsymbol{a})$. The learned solution operator can be further refined to compute the solution for a specific $\boldsymbol{a}^\star$ through continued training only on this parameter (*finetuning*).

In the following, we use the FNO-PINO architecture to learn not a mapping to the PDE solution directly, but to the unknown functions in the *solution structure* (Def. 2.1). With slight abuse of notation we will refer to the output of the neural networks still as $\mathcal{G}_{\boldsymbol{\theta}}(\boldsymbol{a})$. Our approaches to enforce boundary conditions can be used for either PINN or PINO. To illustrate applicability to PINNs, in addition to the regular training and finetuning of the PINO, we consider *PINN-like training*, i.e. learning the solution structure for a specific parameter $\boldsymbol{a}^\star$ without training the solution operator first. This approach is essentially a PINN that uses the FNO architecture instead of dense layers. As we focus on boundary conditions, we do not consider data mismatch terms in the loss, but only physics losses. There are three ways to ensure that the trained solution $\boldsymbol{u}$ satisfies the boundary condition (2).

**Weak boundary conditions.**   Here, the neural operator outputs the solution directly, that is $\boldsymbol{u} = \mathcal{G}_{\boldsymbol{\theta}}(\boldsymbol{a})$, and satisfying the boundary conditions has to be learned during training. This corresponds to

solving the minimization problem

$$\min_{\boldsymbol{\theta}} \sum_{\boldsymbol{a} \in \mathcal{A}_{\text{train}}} \left( \int_{\Omega} \mathcal{P}((\mathcal{G}_{\boldsymbol{\theta}}(\boldsymbol{a}))(\boldsymbol{x}), \boldsymbol{a}(\boldsymbol{x}))^2 d\boldsymbol{x} + \int_{\partial\Omega} \mathcal{B}((\mathcal{G}_{\boldsymbol{\theta}}(\boldsymbol{a}))(\boldsymbol{x}), \boldsymbol{a}(\boldsymbol{x}))^2 dS(\boldsymbol{x}) \right). \quad (3)$$

**Exact boundary conditions.** Here we construct a solution structure in the sense of Definition 2.1 and train the functions $\Psi_i$ such that the parameters satisfy

$$\min_{\boldsymbol{\theta}} \sum_{\boldsymbol{a} \in \mathcal{A}_{\text{train}}} \int_{\Omega} \mathcal{P}(\tilde{\boldsymbol{u}}(\mathcal{G}_{\boldsymbol{\theta}}(\boldsymbol{a}))(\boldsymbol{x}), \boldsymbol{a}(\boldsymbol{x}))^2 d\boldsymbol{x}. \quad (4)$$

for the resulting $\tilde{\boldsymbol{u}}(\Psi_1, \ldots, \Psi_I)$.

**Semi-weak / semi-exact boundary conditions.** Lastly we consider a solution structure that satisfies some boundary conditions but not all. Let $\tilde{\mathcal{B}}$ represent the boundary conditions that $\tilde{\boldsymbol{u}}(\Psi_1, \ldots, \Psi_I)$ does not autoamtically satisfy. The minimization problem solved in training is then posed as a combination of the two previous ones

$$\min_{\boldsymbol{\theta}} \sum_{\boldsymbol{a} \in \mathcal{A}_{\text{train}}} \left( \int_{\Omega} \mathcal{P}(\tilde{\boldsymbol{u}}(\mathcal{G}_{\boldsymbol{\theta}}(\boldsymbol{a}))(\boldsymbol{x}), \boldsymbol{a}(\boldsymbol{x}))^2 d\boldsymbol{x} + \int_{\partial\Omega} \tilde{\mathcal{B}}((\mathcal{G}_{\boldsymbol{\theta}}(\boldsymbol{a}))(\boldsymbol{x}), \boldsymbol{a}(\boldsymbol{x}))^2 dS(\boldsymbol{x}) \right). \quad (5)$$

We use this approach to enforce Dirichlet boundary conditions exactly and Robin conditions weakly.

## 2.1 Exact boundary conditions for scalar differential equations

Starting from Sukumar, et al. (2021)'s approach, the solution consists of two parts, the transfinite interpolant by Rvachev, et al. (2001) for the boundary and a remainder term in the domain

$$u(\boldsymbol{x}) = \underbrace{\sum_{i=1}^{M} w_i(\boldsymbol{x}) u_i(\boldsymbol{x})}_{\text{boundary}} + \underbrace{\Psi(\boldsymbol{x}) \prod_{i=1}^{M} \phi_i(\boldsymbol{x})^{\mu_i}}_{\text{domain}}, \quad (6)$$

$$\forall i \in \{1, \ldots, M\}: \quad w_i(\boldsymbol{x}) = \frac{\prod_{j=1, j \neq i}^{M} \phi_j(\boldsymbol{x})^{\mu_j}}{\sum_{k=1}^{M} \prod_{j=1, j \neq i}^{M} \phi_j(\boldsymbol{x})^{\mu_j}}, \quad (7)$$

where $\phi_i$ is the distance function to $\Gamma_i$. We set $\mu_i = 1$ if $i \in I_D$ and $\mu_i = 2$ if $i \in I_R$, where $I_D \cup I_R = \{1, \ldots, M\}$ are index sets indicating on which segments Dirichlet or Robin boundary conditions are prescribed.

Sukumar, et al. (2021) require an approximate distance function $\phi_i$ in (6), (7) that is both a distance function to $\Gamma_i$ in the sense of Definition 1.1 and normalized with respect to $\Gamma_i$ in the sense of Definition 1.2. However, it is not always possible to find such a function $\phi_i$ that is $C^1$ everywhere. Therefore, we propose to use two different functions instead. The function in (6) and (7), which we still denote as $\phi_i$, is only required to be a distance function to $\Gamma_i$. The function in the local solution structures $u_i$, which we now denote as $\bar{\phi}_i$, only needs to be a normalized function with respect to $\Gamma_i$. Below, we show two ways to choose the local solution structures $u_i$. For comparison, (35) in Appendix A shows Sukumar, et al. (2021)'s choice.

### 2.1.1 Generalized local solution structure (GLSS) for piecewise $C^1$ boundary

For pairwise disjoint boundary segments $\Gamma_1, \ldots, \Gamma_M$ we use the local solution structure

$$\forall i \in I_D: \quad u_i(\boldsymbol{x}) = \begin{cases} g_i(\boldsymbol{x}) + \bar{\phi}_i(\boldsymbol{x}) \tilde{\Psi}_i(\boldsymbol{x}), & \phi_i \text{ has a vanishing gradient,} \\ g_i(\boldsymbol{x}), & \text{else,} \end{cases} \quad (8)$$

$$\forall i \in I_R: \quad u_i(\boldsymbol{x}) = \Psi_i(\boldsymbol{x}) - \bar{\phi}_i(\boldsymbol{x}) \nabla \bar{\phi}_i(\boldsymbol{x}) \cdot \nabla \Psi_i(\boldsymbol{x}) + \bar{\phi}_i(\boldsymbol{x}) (c_i(\boldsymbol{x}) \Psi_i(\boldsymbol{x}) - h_i(\boldsymbol{x})), \quad (9)$$

where the $\bar{\phi}_i$ are normalized with respect to $\Gamma_i$ and the $\tilde{\Psi}_i$ and $\Psi_i$ are unknown functions to be learned. The difference between (8) and (35) is the term $\bar{\phi}_i(\boldsymbol{x}) \tilde{\Psi}_i(\boldsymbol{x})$. Without it, if $\phi_i$ has a vanishing gradient, we would prescribe both $u_i = g_i$ on $\Gamma_i$ and $\frac{\partial u}{\partial \boldsymbol{n}} = \frac{\partial g_i}{\partial \boldsymbol{n}}$ on $\Gamma_i$, which would overdetermine

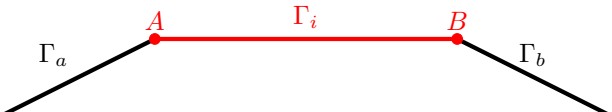

Figure 2: Sketch of intersecting boundary segments.

the problem. The additional term $\bar{\phi}_i(\boldsymbol{x})\tilde{\Psi}_i(\boldsymbol{x})$ avoids that by introducing another unknown function to be trained. In Sukumar, et al. (2021)'s approach, the $\phi_i$ were required to be normalized with respect to $\Gamma_i$ and thus could not have a vanishing gradient, therefore this problem did not arise.

Intersecting boundary segments as sketched in Figure 2 require further modifications to (9), see Appendix B for the reasons. Let $\Gamma_i$ be a boundary segment with neighbors $\Gamma_a$ and $\Gamma_b$, where $A$ and $B$ are the intersection points. If $\Gamma_i$ is a Robin segment, we define the function $\Psi_i$ as

$$\Psi_i(\boldsymbol{x}) = \frac{\phi_B(\boldsymbol{x})}{\phi_A(\boldsymbol{x}) + \phi_B(\boldsymbol{x})} u_A(\boldsymbol{x}) + \frac{\phi_A(\boldsymbol{x})}{\phi_A(\boldsymbol{x}) + \phi_B(\boldsymbol{x})} u_B(\boldsymbol{x}) + \phi_A(\boldsymbol{x})\phi_B(\boldsymbol{x})\bar{\Psi}_i(\boldsymbol{x}), \quad (10)$$

where $\phi_A$ and $\phi_B$ are distance functions to $A$ and $B$. The functions $u_A$ and $u_B$ have to be choosen according to type of boundary conditions prescribed on $\Gamma_a$ and $\Gamma_b$. If $\Gamma_a$ or $\Gamma_b$ is a Dirichlet segment, we set $u_A = g_a$ or $u_B = g_b$. In case of a Robin segment, we introduce a new unknown function $\Psi_A$ or $\Psi_B$ and define $u_A = \Psi_A$ or $u_B = \Psi_B$. The term $\phi_A(\boldsymbol{x})\phi_B(\boldsymbol{x})\bar{\Psi}_i(\boldsymbol{x})$ only needs to be included if both $\Gamma_a$ and $\Gamma_b$ are Dirichlet segments. Function $\bar{\Psi}_i$ is another unknown to be approximated. We show the algorithm for determining local solution structures in the Appendix C].

### 2.1.2 Orthogonal projections (OP)

If all boundary sections with Robin conditions lie in hyperplanes, i.e., for every $\boldsymbol{x} \in \Gamma_i$ the normal vectors $\boldsymbol{n}(\boldsymbol{x})$ are identical, we can use a simpler approach. We choose the normalized functions $\bar{\phi}_i$ to be the exact signed distance function to the hyperplane containing $\Gamma_i$. The local solution structures for Robin conditions are set to

$$\forall i \in I_R: \quad u_i(\boldsymbol{x}) = \underbrace{\Psi_i(\mathcal{N}(\boldsymbol{x};\bar{\phi}_i))}_{\text{boundary value}} + \underbrace{\bar{\phi}_i(\boldsymbol{x})f_i(\mathcal{N}(\boldsymbol{x};\bar{\phi}_i))}_{\text{boundary derivative}}, \quad (11)$$

$$\forall i \in I_R: \quad f_i(\boldsymbol{x}) = c_i(\boldsymbol{x})\Psi_i(\boldsymbol{x}) - h_i(\boldsymbol{x}) \quad \text{with} \quad \mathcal{N}(\boldsymbol{x};\bar{\phi}) := \boldsymbol{x} - \bar{\phi}(\boldsymbol{x})\nabla\bar{\phi}(\boldsymbol{x}) \quad (12)$$

The ansatz uses as a generalized Taylor polynomial expansion by Rvachev, et al. (1995). Here, $\Psi_i$ represents the value and $f_i$ the normal derivative of $u_i$ on $\Gamma_i$. The concatenations $\Psi_i \circ \mathcal{N}(\,\cdot\,;\bar{\phi}_i)$ and $f_i \circ \mathcal{N}(\,\cdot\,;\bar{\phi}_i)$ are the so called normalizers of $\Psi_i$ and $f_i$, see Appendix D.

The first term in (11) determines the value on $\Gamma_i$ but has zero derivative whilst the second term is zero on $\Gamma_i$ but has non-zero derivative equal to the Robin condition. Because $\bar{\phi}_i$ is the exact signed distance function, $\mathcal{N}(\,\cdot\,;\bar{\phi}_i)$ is an orthogonal projection mapping its argument onto the corresponding hyperplane. Therefore, each $\Psi_i$ is only evaluated on its corresponding hyperplane and we can set

$$\forall i \in I_R: \quad \Psi_i(\boldsymbol{x}) = g(\boldsymbol{x}) + \tilde{\Psi}(\boldsymbol{x}) \prod_{k \in I_D} \phi_k(\boldsymbol{x}), \quad (13)$$

where $g$ is a function satisfying all Dirichlet conditions. This avoids discontinuities of the transfinite interpolant at intersection points since only the single function $\tilde{\Psi}$ needs to be trained.

### 2.2 Exact boundary conditions for systems of partial differential equations

Consider a system of differential equations with solution $\boldsymbol{u} : \mathbb{R}^2 \supset \Omega \to \mathbb{R}^n$ and boundary conditions prescribed on segments $\Gamma_1, \ldots, \Gamma_M \subset \partial\Omega$ with each $\Gamma_i$ being $C^1$. For $i \in \{1, \ldots, M\}$ and $\boldsymbol{x} \in \Gamma_i$ let $\boldsymbol{b}_i^{(1)}(\boldsymbol{x}), \ldots, \boldsymbol{b}_i^{(n)}(\boldsymbol{x})$ be a basis of $\mathbb{R}^n$. Let $IJ_D, IJ_R \subset I \times J := \{1, \ldots, M\} \times \{1, \ldots, n\}$ be index sets such that

$$\forall (i,j) \in IJ_D : \forall \boldsymbol{x} \in \Gamma_i : \qquad\qquad \boldsymbol{b}_i^{(j)}(\boldsymbol{x}) \cdot \boldsymbol{u}(\boldsymbol{x}) = g_i^{(j)}(\boldsymbol{x}), \quad (14)$$

$$\forall (i,j) \in IJ_R : \forall \boldsymbol{x} \in \Gamma_i : \qquad \frac{\partial\left(\boldsymbol{b}_i^{(j)}(\boldsymbol{x}) \cdot \boldsymbol{u}(\boldsymbol{x})\right)}{\partial\boldsymbol{n}} + \boldsymbol{c}_i^{(j)}(\boldsymbol{x}) \cdot \boldsymbol{u}(\boldsymbol{x}) = h_i^{(j)}(\boldsymbol{x}), \quad (15)$$

Without loss of generality, we assume that for every $(i,j) \in IJ_R$ and $\boldsymbol{x} \in \Gamma_i$ the $c_i^{(j)}(\boldsymbol{x})$ lies in $\text{span}(\{\boldsymbol{b}_i^{(k)}(\boldsymbol{x}) \,|\, (i,k) \notin IJ_D\})$. The generic solution structure now becomes

$$\boldsymbol{u}(\boldsymbol{x}) = \sum_{i=1}^{M} w_i(\boldsymbol{x})\boldsymbol{u}_i(\boldsymbol{x}) \;+\; [\Psi^{(1)}(\boldsymbol{x}), \ldots, \Psi^{(n)}(\boldsymbol{x})]^T \prod_{i=1}^{M} \phi_i(\boldsymbol{x})^{\mu_i}, \tag{16}$$

$$\mu_i = \begin{cases} 2, & \exists j : (i,j) \in IJ_R, \\ 1, & \text{else}, \end{cases} \tag{17}$$

with weights given in (7). Functions $\boldsymbol{u}_i$ are expressed as a linear combination of the basis functions

$$\forall i \in I : \quad \boldsymbol{u}_i(\boldsymbol{x}) = \sum_{j=1}^{n} \boldsymbol{b}_i^{(j)}(\boldsymbol{x}) u_i^{(j)}(\boldsymbol{x}). \tag{18}$$

**Proposition 2.2.** If the vectors $\boldsymbol{b}_i^{(1)}(\boldsymbol{x}), \ldots, \boldsymbol{b}_i^{(n)}(\boldsymbol{x})$ form a basis of $\mathbb{R}^n$ for every $\boldsymbol{x} \in N(\Gamma_i)$ where $N(\Gamma_i)$ is an open neighborhood of $\Gamma_i$, we have

$$\forall (i,j) \in I \times J : \qquad\qquad \forall \boldsymbol{x} \in \Gamma_i : \qquad \boldsymbol{b}_i^{(j)}(\boldsymbol{x}) \cdot \boldsymbol{u}(\boldsymbol{x}) = u_i^{(j)}(\boldsymbol{x}), \tag{19}$$

$$\forall (i,j) \in I \times J \; \begin{smallmatrix}\text{with } \phi_i^{\mu_i} \text{ having} \\ \text{a vanishing gradient}\end{smallmatrix} : \quad \forall \boldsymbol{x} \in \Gamma_i : \quad \frac{\partial\left(\boldsymbol{b}_i^{(j)}(\boldsymbol{x}) \cdot \boldsymbol{u}(\boldsymbol{x})\right)}{\partial \boldsymbol{n}} = \frac{\partial u_i^{(j)}(\boldsymbol{x})}{\partial \boldsymbol{n}}. \tag{20}$$

*Proof.* The proof is shown in Appendix E. □

Therefore, every $u_i^{(j)}$ has to satisfy the corresponding boundary condition. For Dirichlet conditions this is achieved by setting

$$\forall (i,j) \in IJ_D : \quad u_i^{(j)}(\boldsymbol{x}) = \begin{cases} g_i^{(j)}(\boldsymbol{x}) + \bar{\phi}_i(\boldsymbol{x})\tilde{\Psi}_i^{(j)}(\boldsymbol{x}), & \phi_i^{\mu_i} \text{ has a vanishing gradient}, \\ g_i^{(j)}(\boldsymbol{x}), & \text{else}, \end{cases} \tag{21}$$

for both GLSS and OP. However, the two approaches differ in their treatment of Robin conditions.

### 2.2.1 GLSS

If all boundary segments $\Gamma_1, \ldots, \Gamma_M$ are pairwise disjoint, we set

$$u_i^{(j)}(\boldsymbol{x}) = \Psi_i^{(j)}(\boldsymbol{x}) - \bar{\phi}_i(\boldsymbol{x})\nabla\bar{\phi}_i(\boldsymbol{x}) \cdot \nabla\Psi_i^{(j)}(\boldsymbol{x}) + \bar{\phi}_i(\boldsymbol{x})f_i^{(j)}(\boldsymbol{x}), \tag{22}$$

$$f_i^{(j)}(\boldsymbol{x}) = \boldsymbol{c}_i^{(j)}(\boldsymbol{x}) \cdot \sum_{k=1,(i,k)\notin IJ_D}^{n} \boldsymbol{b}_i^{(k)}(\boldsymbol{x})\Psi_i^{(k)}(\boldsymbol{x}) \;-\; h_i^{(j)}(\boldsymbol{x}). \tag{23}$$

for $(i,j) \in IJ_R$. For $(i,j) \in I \times J \setminus (IJ_D \cup IJ_R)$, we define $u_i^{(j)}(\boldsymbol{x}) = \Psi_i^{(j)}(\boldsymbol{x})$. As above, the functions $\Psi_i^{(j)}$ have to be modified, if $\Gamma_i$ has intersection points with other segments.

**Intersecting boundary segments.** We generalize our approach to the system case and let

$$\Psi_i^{(j)}(\boldsymbol{x}) = \boldsymbol{b}_i^{(j)}(\boldsymbol{x}) \cdot \left( \frac{\phi_B(\boldsymbol{x})}{\phi_A(\boldsymbol{x}) + \phi_B(\boldsymbol{x})} \boldsymbol{u}_A(\boldsymbol{x}) + \frac{\phi_A(\boldsymbol{x})}{\phi_A(\boldsymbol{x}) + \phi_B(\boldsymbol{x})} \boldsymbol{u}_B(\boldsymbol{x}) \right) + \phi_A(\boldsymbol{x})\phi_B(\boldsymbol{x})\bar{\Psi}_i^{(j)}(\boldsymbol{x}). \tag{24}$$

The construction of the functions $\boldsymbol{u}_A$ and $\boldsymbol{u}_B$ is more difficult than in the scalar case. We demonstrate how to do this with an example. Consider two segments $\Gamma_1$ and $\Gamma_2$ with intersection point $P$ and assume for simplicity that $u \in \mathbb{R}^3$. Let Dirichlet conditions be prescribed on $\Gamma_1$ with respect to the basis vectors $\boldsymbol{b}_1^{(1)}(P) = (1,0,0)^T$ and $\boldsymbol{b}_1^{(2)}(P) = (0,1,0)^T$ and on $\Gamma_2$ with respect to the vector $\boldsymbol{b}_2^{(1)}(P) = (1,1,0)^T$. These three basis vectors span a two-dimensional subspace with basis $(1,0,0)^T, (0,1,0)^T$. We define $\boldsymbol{u}_P$ as a linear combination of this basis and an unknown component acting on the orthogonal complement, i.e.

$$\boldsymbol{u}_P(\boldsymbol{x}) = g_P^{(1)} \begin{pmatrix} 1 \\ 0 \\ 0 \end{pmatrix} + g_P^{(2)} \begin{pmatrix} 0 \\ 1 \\ 0 \end{pmatrix} + \Psi_P^{(3)}(\boldsymbol{x}) \begin{pmatrix} 0 \\ 0 \\ 1 \end{pmatrix}. \tag{25}$$

Note that the constants $g_P^{(1)}$ and $g_P^{(2)}$ have to be chosen such that $\boldsymbol{u}_P$ satisfies all Dirichlet conditions prescribed in $P$. A complete algorithm can be found in the Appendix C.

Table 1: Size, training and inference times of the FNO for the four different approaches to enforce boundary conditions for the Darcy flow and Navier-Stokes equations.

| | Trainable parameters | Checkpoint size (MByte) | Training time (min) | Inference time (sec) |
|---|---|---|---|---|
| | | Darcy flow | | |
| GLSS | 13,132,932 | 105 | 182.35 | 0.0130 |
| OP | 13,132,674 | 105 | 181.68 | 0.0120 |
| Semi-weak | 13,132,545 | 105 | 180.79 | 0.0104 |
| Weak | 13,132,545 | 105 | 181.44 | 0.0101 |
| | | Navier-Stokes equations | | |
| GLSS | 13,133,706 | 105 | 9.79 | 0.0298 |
| OP | 13,133,190 | 105 | 9.76 | 0.0285 |
| Semi-weak | 13,132,803 | 105 | 9.49 | 0.0248 |
| Weak | 13,132,803 | 105 | 9.62 | 0.0238 |

### 2.2.2 OP

If all boundary segments $\Gamma_i$ lie in hyperplanes and $\boldsymbol{b}^{(j)} := \boldsymbol{b}_1^{(j)} = \cdots = \boldsymbol{b}_M^{(j)}$ holds for every $j = 1, \ldots, n$, the global solution structure, given by (16) and (18), simplifies to

$$\boldsymbol{u}(\boldsymbol{x}) = \sum_{j=1}^{n} \boldsymbol{b}^{(j)}(\boldsymbol{x}) \sum_{i=1}^{M} w_i(\boldsymbol{x}) u_i^{(j)}(\boldsymbol{x}) \quad + \quad [\Psi^{(1)}(\boldsymbol{x}), \ldots, \Psi^{(n)}(\boldsymbol{x})]^T \prod_{i=1}^{M} \phi_i(\boldsymbol{x})^{\mu_i}. \quad (26)$$

We choose the local solution structures as

$$u_i^{(j)}(\boldsymbol{x}) = \begin{cases} \begin{cases} g_i^{(j)}(\boldsymbol{x}) + \bar{\phi}_i(\boldsymbol{x}) \tilde{\Psi}_i(\boldsymbol{x}), & \phi_i \text{ has a vanishing gradient} \\ g_i^{(j)}(\boldsymbol{x}), & \text{else} \end{cases}, & (i,j) \in IJ_D, \\ \bar{\Psi}^{(j)}(\mathcal{N}(\boldsymbol{x}; \bar{\phi}_i)) + \bar{\phi}_i(\boldsymbol{x}) f_i^{(j)}(\mathcal{N}(\boldsymbol{x}; \bar{\phi}_i)), & (i,j) \in IJ_R, \\ \bar{\Psi}^{(j)}(\boldsymbol{x}), & \text{else} \end{cases} \quad (27)$$

The functions $f_i^{(j)}$ are defined as

$$f_i^{(j)}(\boldsymbol{x}) = \boldsymbol{c}_i^{(j)}(\boldsymbol{x}) \cdot \sum_{k=1, (i,k) \notin IJ_D}^{n} \bar{\Psi}^{(k)}(\boldsymbol{x}) \boldsymbol{b}^{(k)}(\boldsymbol{x}) \quad - \quad h_i^{(j)}(\boldsymbol{x}), \quad (28)$$

and the $\bar{\Psi}^{(j)}$ are defined as

$$\bar{\Psi}^{(j)}(\boldsymbol{x}) = g^{(j)}(\boldsymbol{x}) + \tilde{\Psi}^{(j)}(\boldsymbol{x}) \prod_{i=1, (i,j) \in IJ_D}^{M} \phi_i(\boldsymbol{x}). \quad (29)$$

Each function $g^{(j)}$ is chosen in a way that it satisfies all Dirichlet conditions prescribed with respect to the basis vector $\boldsymbol{b}^{(j)}$ and $\tilde{\Psi}^{(j)}$ is an unknown function to be approximated.

**Theorem 2.3.** The derived solution structure satisfies the boundary conditions (14) and (15) for both the GLSS and OP approach.

*Proof.* The proof is in Appendix F. □

## 3 Numerical results

Details regarding the architecture and training of the network can be found in Appendix G. Table 1 shows different training-related parameters of the networks arising from the four approaches for the two benchmarks. Because the size of the network is dominated by the size of the four convolution layers, the number of trainable parameters varies only very slightly. There is no discernible impact on the size of the checkpoint files. Training times are stable, with semi-weak boundary conditions training the fastest in both cases but the difference to the slowest GLSS is below $3\%$. Inference times increase for GLSS and OP compared to weakly enforced boundary conditions. We see the largest increase by about 29% for GLSS for the Darcy flow. Note that training for the Navier-Stokes equations is much faster because we train only solutions and no solution operator.

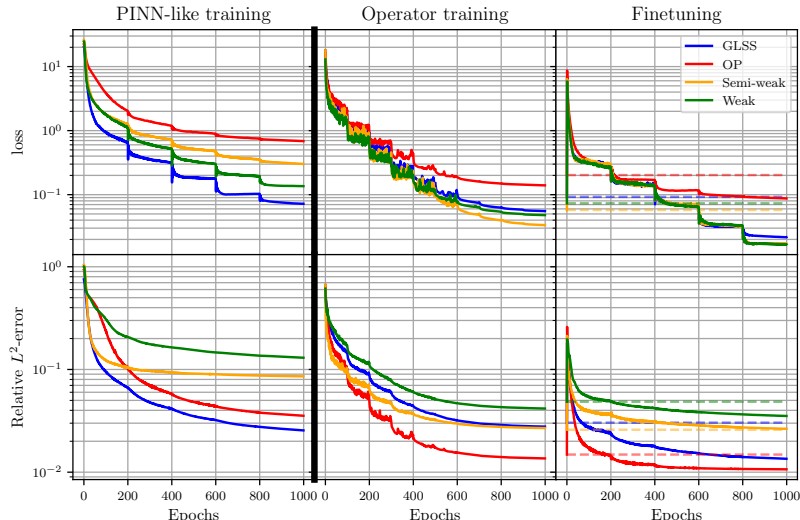

Figure 3: Training progress and errors on the validation set for different ways to enforce boundary conditions for the Darcy flow.

## 3.1 Darcy Flow

As scalar test problem, we consider the Darcy flow equation governing fluid flow in porous media Darcy (1856). The detailed numerical setup can be found in Appendix H, including the precise local solution structures used for GLSS and OP. Figure 3 shows training loss (upper) and validation error (lower). The left column shows the loss and error curve for the PINN-like training. Both these curves are the average loss and $l^2$-error over the 100 parameters that the PINO was trained on individually in PINN-style. The middle column shows the loss and error curve for PINO trained on 400 parameters. The right column shows the loss and error curve for finetuning, Dotted lines indicate the loss and error value at the very beginning of the finetuning.

All cases train reasonably well, reducing the loss function by at least one order of magnitude (OP for PINN-like training) and two or more orders in most cases. Losses are not indicative of achieved validation errors. For the PINN-style training, OP and GLSS are more accurate than weak or semi-weak boundary conditions. The same holds true for finetuning, where OP is slightly more accurate than GLSS. For operator training, OP is more accurate than GLSS which performs on par with semi-weak and better than weak boundary conditions. In summary, for the Darcy flow, even though losses do not necessarily decay faster, OP and GLSS in almost all cases produces more accurate solutions than weak or semi-weak boundary conditions. Table 2 shows the average $l_2$-error plus standard deviation (left column), best case $l_2$-error (middle column) and worst case $l_2$-error (right column). For operator training and finetuning, OP is the most accurate approach whilst GLSS is the most accurate for PINN-like training. For best case errors, shown in the middle column, there is no clear benefit from the two new approaches However, there are substantial gains in accuracy from OP and GLSS for the worst case in PINN-like training and finetuning and from OP in operator training. Plots of the median, best- and worst-case solutions can be found in Appendix I.

Table 2: $l_2$-errors of the predicted $u$ against the analytical solution for the four different approaches to enforce boundary conditions for the Darcy flow problem.

|  | Operator training | | | Finetuning | | | PINN-like training | | |
|---|---|---|---|---|---|---|---|---|---|
|  | Average | Best | Worst | Average | Best | Worst | Average | Best | Worst |
| GLSS | 0.03±0.04 | 0.004 | 0.27 | 0.01±0.01 | 0.003 | 0.06 | 0.02±0.02 | 0.005 | 0.08 |
| OP | 0.02±0.01 | 0.003 | 0.06 | 0.01±0.01 | 0.002 | 0.04 | 0.04±0.02 | 0.006 | 0.11 |
| S-Weak | 0.03±0.03 | 0.002 | 0.17 | 0.03±0.02 | 0.003 | 0.10 | 0.09±0.03 | 0.021 | 0.17 |
| Weak | 0.05±0.05 | 0.008 | 0.28 | 0.04±0.05 | 0.003 | 0.26 | 0.13±0.14 | 0.008 | 0.64 |

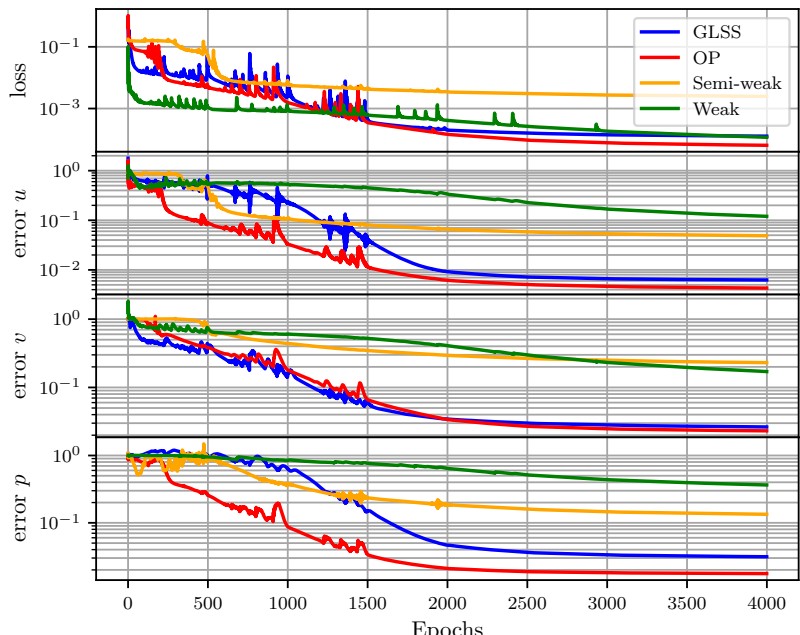

Figure 4: Training loss (upper) and validation error in the $u$ (upper middle) and $v$ (lower middle) velocity component and pressure $p$ (lower) for the Navier-Stokes equations.

## 3.2 Navier-Stokes equations

We use the standard benchmark by Turek, et al. (1996), simulating 2D stationary flow through a channel and across a cylinder. The details of the problem setup can be found in Appendix J, including the precise form of the local solution structures used for GLSS and OP.

Figure 4 shows the training losses in the upper figure and the errors in the velocity components $u$ and $v$ and the pressure $p$ against the numerically computed reference solution. After 4000 epochs, losses for GLSS, OP and weak boundary conditions are similar but the loss for semi-weak remains somewhat higher. In terms of errors, we again see a clear benefit in terms of accuracy from GLSS and OP as they outperform weak and semi-weak boundary conditions in all three solution components.

To further assess accuracy we consider three practically relevant diagnostic quantities: pressure difference, drag coefficient and lift coefficient, see Turek, et al. (1996) for their definition. Table 3 shows the values computed from the PINO using the four different ways to enforce boundary conditions and, in brackets, the relative error against the reference values by Nabh (1998). We again see a noticeable increase in accuracy from GLSS and OP over weak or semi-weak boundary conditions. Pressure difference and drag coefficient are predicted with high accuracy. While relative errors for the lift coefficient are large, they are still orders of magnitude smaller than for the weak or semi-weak approach.

Table 3: Physically important parameters computed from the Navier-Stokes solution. The reference values are provided by Nabh (1998) with 9 digit accuracy and we rounded them to 4 digits. The relative error against those reference values is shown in brackets.

|  | Pressure difference | Drag coefficient | Lift coefficient |
|---|---|---|---|
| GLSS | 0.1150 ( 2.1%) | 5.5336 ( 0.8%) | -0.0058 ( 155%) |
| OP | 0.1145 ( 2.6%) | 5.5366 ( 0.8%) | 0.0024 ( 77%) |
| Semi-weak | 0.0678 (42.3%) | 3.8221 (31.5%) | -0.3759 (3646%) |
| Weak | 0.0902 (23.2%). | 4.6633 (16.4%) | 0.3849 (3531%) |
| Reference values | 0.1175 | 5.5795 | 0.0106 |

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
