# OpenReview forum: "Enforcing boundary conditions for physics-informed neural operators"
_NeurIPS.cc/2025/Conference — Submitted to NeurIPS 2025_

### Official Review · Reviewer_GieS · 2025-06-09

**Clarity:** 3
**Significance:** 2
**Originality:** 2
**Rating:** 3
**Confidence:** 3

**Summary:**

This paper introduces two improved approaches to enforce boundary conditions on piecewise $C^1$ boundaries in physics informed neural operator: generalized local solution structure (GLSS) and orthogonal projection (OP). Both of the proposed methods can make sure that the proposed solution satisfies boundary conditions by construction. GLSS makes use of distance function and normalized function to construct solutions, loosening the requirement of distance function to be normalized in a previous work. Once the boundaries with Robin boundary conditions are in the same hyperplane, the form of solution can be constructed differently to reduce the number of unknown functions to be trained. This method is called orthogonal projection (OP). Both of these methods can be applied to solve both single PDEs and PDE systems. These two methods are verified on Darcy flow equation and Navier Stokes equations with numerical experiments.

**Questions:**

1. Why is FNO chosen as the framework for implementation in this work? It is not very suitable for irregular domains involved here and the computation of derivatives can only be approximated with finite difference.
2. Table 1 shows the convergence time of different methods for Darcy and NS problems. Why is the convergence of NS much faster than that of Darcy? NS is a non-linear problem and is supposed to be harder to solve.

**Ethical Concerns:**

["NO or VERY MINOR ethics concerns only"]

**Final Justification:**

A major reorganization of the materials in this paper is required before this paper can be accepted, which could make this submission a totally different one.

**Limitations:**

Yes, the limitations are discussed in this manuscript.

**Paper Formatting Concerns:**

1. A click on one citation item in the main text is expected to redirect the reader to the cited item in the reference list. This property is not realized in this manuscript.
2. A conclusion section is lacked in this manuscript. The main text ends abruptly after a discussion on NS equations.

**Quality:**

3

**Strengths And Weaknesses:**

Strengths:
1. The motivation of this paper is thoroughly discussed.
2. The proposed methods, though complicated, are stated in a very clear manner.

Weaknesses:
1. There are problems with the formatting of this paper. Please refer to formatting section of the review for more details.
2. The title of this paper mentions 'physics informed neural operator', but it's not about operator learning. Instead, it is about PINN, which requires retraining for every new case.
3. Only two equations (Darcy flow and NS) are involved in the numerical experiments in this paper, which is not sufficient to show the ability of the proposed methods to solve different problems.
4. Related to 2, if the training time for a 2D Darcy is as long as 180 minutes, and a retraining is required for every new case, then this method is not very useful in practical use cases.

---

> ### Author Rebuttal · Authors · 2025-07-30
>
> ## Weaknesses
>
> > There are problems with the formatting of this paper. Please refer to formatting section of the review for more details.
>
> The citations will be fixed and a conclusion section will be added.
>
> > The title of this paper mentions 'physics informed neural operator', but it's not about operator learning. Instead, it is about PINN, which requires retraining for every new case.
>
> This is misunderstanding. For the Darcy flow, we investigate three "styles" of training, a PINN-like training where we learn a single solution, an operator training that learns the mapping from the parameter pair $(\alpha_i, \beta_i)$ controlling the source term $f(x,y)$ to the solution and a fine-tuning, where the operator network is used as the starting point for learning a specific solution. This might have been confusing in the paper, since only single solutions are learned for the Navier-Stokes equations. We will augment the results for the NSE by also training an operator, see reply below, which will also provide stronger evidence for the efficacy of our approach.
>
> > Only two equations (Darcy flow and NS) are involved in the numerical experiments in this paper, which is not sufficient to show the ability of the proposed methods to solve different problems.
>
> As in the reply to reviewer [nPRe](https://openreview.net/forum?id=yT8hGWqoBm&noteId=Dfl8DpB7x0), we would argue that the Navier-Stokes equations are a complex example and encompass other equations as special cases and, together with the Darcy flow problem, provides strong evidence that the new approach can be useful.
>
> In preparation for this rebuttal, we have started benchmarking the approach for a third type of practically relevant PDE, the Helmholtz equation. Preliminary results posted in the previously linked reply confirm the efficacy of our approach and a full set of results will be included in the revised paper.
>
> > Related to 2, if the training time for a 2D Darcy is as long as 180 minutes, and a retraining is required for every new case, then this method is not very useful in practical use cases.
>
> This is a misunderstanding, see also the reply above: for Darcy, we learn the operator that maps from the parameters controlling the source term to the solution. Therefore, after training, it can produce solutions for a wide range of parameters without retraining with a single inference. This was, however, not very clear from the headings in Table 1 as the Darcy flow trains an operator while for the NSE we trained only a single solution. We will augment the table by also reporting training times for PINN-style training of a single solution of Darcy flow.
>
> **Darcy flow (PINN-style training)**
> | BC type   | Training time |
> | --------- | ------ |
> | GLSS      | 85.6s  |
> | OP        | 85.8s  |
> | Semi-weak | 82.6s  |
> | Weak      | 82.9s  |
>
> Furthermore, we are currently producing results for operator training for the Navier-Stokes equations, learning the mapping $\nu \mapsto \mathbf{u}$, that is, the operator that  maps the fluid viscosity to the solution.
> We trained on 80 solutions for values of $5 \times 10^{-4} < \nu < 0.1$ and evaluated performance of the operator on 20 untrained viscosity values in this range. Average errors are reported in the table below
>
> **NSE (operator training)**
> | BC type   | Average error $u$ | Average error $v$ | Average error $p$ |
> | --------- | ----------------- | ----------------- | ----------------- |
> | GLSS      | 0.069             | 0.029             | 0.066             |
> | OP        | 0.025             | 0.033             | 0.078             |
> | Weak      | 0.182             | 0.162             | 0.232             |
>
> We do not yet have results for the semi-weak case to report but those will also be added to the revised paper. Note that our approaches (GLSS and OP) again increase accuracy by about one order of magnitude compared to weakly enforced boundary conditions.
>
> ## Questions:
>
> > Why is FNO chosen as the framework for implementation in this work? It is not very suitable for irregular domains involved here and the computation of derivatives can only be approximated with finite difference.
>
> FNO seems to be an established approach towards learning PDE solution operators, and widely used. We agree with the reviewer that it is not by itself suitable for irregular domains. As described in Appendix G, for the numerical experiments we used a bounding box as the computational domain. There are extensions of FNO available for irregular domains, e.g., Liu et al., *Domain agnostic fourier neural operators*, NeurIPS 2023, or Li et al., *Fourier Neural Operator with Learned Deformations for PDEs on General Geometries*, JMLR 24(388), pp 1--26, 2023.
>
> However, our approach does not require the use of FNOs to learn the unknown functions. Other neural operators like the multiwavelet-based neural operator (Gupta, et al., *Multiwavelet-based Operator Learning for Differential Equations*, NeurIPS 2021) or DeepONets (Lu et al., *Learning nonlinear operators via DeepONet based on the universal approximation theorem of operators*. Nature Machine Intelligence, 3, 218-229, 2021)/physics-informed DeepONets (Wang et al., *Learning the solution operator of parametric partial differential equations with physics-informed DeepONets*, Science Advances 7(40), 2021) could be used. This was probably not pointed out clearly enough in the paper.
>
> We will state this explicitly directly after Definition 2.1, together with an example, see comments by reviewer [TWYM](https://openreview.net/forum?id=yT8hGWqoBm&noteId=FZiqgE7ZyM).
>
> > Table 1 shows the convergence time of different methods for Darcy and NS problems. Why is the convergence of NS much faster than that of Darcy? NS is a non-linear problem and is supposed to be harder to solve.
>
> Please see the reply above - the reason is that in case of the NSE, only a single solution is learned whilst for Darcy flow the network learns the solution operator. As described, we will unify this and also learn a solution operator for the NSE.

---

> > ### Comment · Reviewer_GieS · 2025-08-05
> > **Reply to Rebuttal**
> >
> > The reviewer would like to thank the authors for their detailed explanation to the questions raised. Their rebuttal does help me understand the contents of this submission better. I would reconsider my original ratings. However, in the current form, as a major reorganization of the materials would be required, which could make this paper a totally different one, I would not recommend acceptance for this submission.

---

> > > ### Author Response · Authors · 2025-08-06
> > >
> > > > However, in the current form, as a major reorganization of the materials would be required, which could make this paper a totally different one, I would not recommend acceptance for this submission.
> > >
> > > In essence, the required changes would be some reordering of the description of the approach and adding the operator-training results for NSE plus results for the Helmholtz equation. We would disagree that would result in a completely different paper.

---

### Official Review · Reviewer_nPRe · 2025-06-30

**Clarity:** 2
**Significance:** 2
**Originality:** 2
**Rating:** 2
**Confidence:** 4

**Summary:**

This work studies enforcing boundary conditions in Neural Operators, which is an important problem but has been studied in other works that are not referenced. The method has u minimize the PDE loss and composes it in terms of orthogonal polynomials and by construction they satisfy the boundary conditions.

**Questions:**

1. Why is the method tested on only stationary problem, e.g., scalar Darcy flow and stationary NS equations? Can it not work with a time component?
2. This basis approach seems similar to finite elements - how does it compare/contrast?
3. What does fine-tuning mean in this context?

**Ethical Concerns:**

["NO or VERY MINOR ethics concerns only"]

**Final Justification:**

I would like to thank the authors very much for their detailed rebuttal. While I agree that the exact boundary conditions differ from those in Saad et al. and the methodology is distinct, Robin BCs are simply a linear combination of Neumann and Dirichlet boundary conditions, which are both covered in that work and could easily be extended to that case. I believe the BOON method should at least be compared to as a baseline.

Also, just because the boundary conditions need to be stationary does not require the PDEs to be stationary and more realistic time-dependent cases can be tested.

In addition, while the constraint type may in different in Hansen et al. imposing conservation laws, it compares a general framework from any linear constraint using an oblique or orthogonal projection. Hence, the orthogonal projection to impose the constraint is the same methodology. In fact, Mouli et al., "Using uncertainty quantification to characterize and improve out-of-domain learning for pdes", ICML 2024 proposes the Operator-ProbConserv, which applies this oblique (more general than orthogonal) projection to impose physical constraints using Neural Operators, which I think is relevant. In addition in Cheng et al., the constraints are also imposed via orthogonal projection. So, I do not think there is much novelty in the orthogonal projection approach.

I appreciate the focus on enforcing hard constraints in this work and believe it could be beneficial to the community with a revised future submission that takes these points into consideration. At this time, I will be maintaining my score.

**Limitations:**

yes

**Quality:**

2

**Strengths And Weaknesses:**

## Strengths
- It is good that the authors target an important and practical problem of enforcing boundary conditions in Neural Operators and that realistic BS including Dirichlet, Neumann and Robin are considered.

## Weaknesses
- A major weakness is that other relevant works that impose boundary conditions as hard constraints in Neural Operators, e.g., Saad et al., "Guiding continuous operator learning through Physics-based boundary constraints", ICLR, 2023 are not referenced or compared to.
- Only two test equations are benchmarked and in addition they are stationary.
- The references are not cited properly with citep for parathesis.
- Unclear the novelities compared to Sukumar, et al. (2021)
- The review of PINNs and PINO could be moved to appendix
- The method is still similar to PINNs where the domain knowledge is incorporated as a soft constraint in the loss function contrary to hard-constrained works, e.g., Hansen et al., "Learning Physical Models that Can Respect Conservation Laws, ICML, 2023, Negiar et al., "Learning differentiable solvers for systems with hard constraints", ICLR, 2023, Saad et al., "Guiding continuous operator learning through Physics-based boundary constraints", ICLR, 2023, Cheng et al., "Gradient-free generation for hard-constrained systems", ICLR, 2025.
- There is no conclusion section

---

> ### Author Rebuttal · Authors · 2025-07-28
>
> ## Summary
>
> > The method has u minimize the PDE loss and composes it in terms of orthogonal polynomials and by construction they satisfy the boundary conditions
>
> While the reviewer is correct that our approach satisfies boundary conditions by construction, we would like to point out that we do not use orthogonal polynomials. The OP approach introduced in Section 2.1.2 uses orthogonal *projections*.
>
> ## Weaknesses
>
> > - A major weakness is that other relevant works that impose boundary conditions as hard constraints in Neural Operators, e.g., Saad et al., "Guiding continuous operator learning through Physics-based boundary constraints", ICLR, 2023 are not referenced or compared to.
>
> We will add and discuss the provided reference in the revised paper. However, please note that the approach by *Saad et al.* is quite different to what we propose: they directly modify the integral kernel in the neural operator to enforce one type of boundary condition (Dirichlet, Neumann or periodic). While their approach can be flexibly applied to different types of neural operators, it requires boundary condition specific modifications of the architecture. It is not clear from the paper how mixed boundary conditions could be enforced, which is one of the major contributions of our work.
>
> We very much agree with the reviewer that a comparison of our against their approach would be interesting but also would argue that, because of the differences in the approaches, this cannot be done realistically within the scope of this paper.
>
> > - Only two test equations are benchmarked and in addition they are stationary.
>
> Since the Navier-Stokes equations are a system of differential equations with complex dynamics that also include several other commonly used benchmarks as special case, we would argue that the two test problems are sufficient to provide a reasonable degree of confidence in the performance of our approach.
>
> However, in preparing this rebuttal, we have started to benchmark our approach also for the Helmholtz equation
> $$\Delta u + k^2 u = 0$$
> on the same L-shaped domain and with the same type of boundary condition as the Darcy flow. Errors are computed against the analytical solution
> $$u(x,y) = \sin\left( \frac{\sqrt{3}}{2} k x \right) \sin\left( \frac{1}{2} k y \right)$$
> with $k=1$.
> Operator training, learning the mapping $k \mapsto u$, and fine-tuning are still running and results will be added to the revised paper but we can already report errors after 2000 epochs of PINN-style training.
>
> **Helmholtz equation (PINN-style training)**
> | BC type | Error |
> | --------- | ------ |
> | GLSS   | 0.014 |
> | OP       | 0.005 |
> | Semi-weak | 0.016 |
> | Weak | 0.603 |
>
> The Helmholtz equation is another type of PDE with great relevance for applications. While preliminary, the results again confirm that our OP approach can deliver substantial improvements in accuracy, particularly compared to only weakly enforced boundary conditions.
>
> > - The references are not cited properly with citep for parathesis.
>
> We used the *citet* command for in-text citations as suggested by the NeurIPS 2025 guidelines. We are, however, happy to change this to whatever style would be requested for a possible camera-ready version.
>
> > - Unclear the novelities compared to Sukumar, et al. (2021)
>
> We tried to spell out clearly in the introduction how our approach extends that by *Sukumar et al.* and fixes issues arising with their approach for problems with non-smooth boundaries. Figure 1 shows the problem with their approach, which can break down completely if $\partial \Omega \notin C^1$. We will rework the paragraph on contributions (line 75 to 84) to point this out more clearly.
>
> We'd like to stress that the requirement in Sukumar, et al. (2021) of $\partial \Omega \in C^1$ is a severe restriction: Many if not most domains in applications do not have globally smooth boundaries; not even fairly simple geometries like squares or rectangles have $C^1$ boundaries (and Sukumar's approach can break down already for those, see Figure 1 in the paper). Therefore, removing this requirement is an important step towards making PINNs or NOs with hard-constrained boundary conditions much more practically useful.
>
> > - The review of PINNs and PINO could be moved to appendix
>
> We will restructure some of the paper also in reponse to the detailed suggestions made by reviewer [TWYM](https://openreview.net/forum?id=yT8hGWqoBm&noteId=FZiqgE7ZyM) and move some parts to the Appendix and some other parts from the Appendix to the main paper. However, given that the difference between operator training and fine-tuning is critical for the reader to understand the results shown later, we would argue that it is worthwhile to keep at the very least the discussion of these concepts.
>
> > - The method is still similar to PINNs where the domain knowledge is incorporated as a soft constraint in the loss function contrary to hard-constrained works, e.g., Hansen et al., "Learning Physical Models that Can Respect Conservation Laws, ICML, 2023, Negiar et al., "Learning differentiable solvers for systems with hard constraints", ICLR, 2023, Saad et al., "Guiding continuous operator learning through Physics-based boundary constraints", ICLR, 2023, Cheng et al., "Gradient-free generation for hard-constrained systems", ICLR, 2025.
>
> Our approach hard-constrains boundary conditions, it does *not* incorporate them into the loss function. Thus our approach *removes* the boundary loss term from PINN or PINO loss functions, and enforces satisfying the boundary conditions by construction. We thank the reviewer for the provided references and will augment our related works section accordingly but we would also like to point out that our approach differs in significant ways from those papers:
>
> - *Hansen et al.* consider conservation laws and use a finite volume method. They enforce conservation via linear probabilistic constraints and use a two-step procedure where first a ML model estimates a predictive distribution to which then a discretization of the integral form of the conservation law is applied to enforce the physical conservation constraint. Boundary conditions are not a focus of this work.
> - *Negiar et al.* add a differentiable constraining layer to the neural network to hard-constrain the PDE-residual. While it is conceivable that boundary conditions could be enforced this way, they are not a focus of this work.
> - *Saad et al.* was discussed above.
> - *Cheng et al.* use extrapolation correction interpolation sampling, a unified gradient-free sampling framework, to guide an unconstrained pre-trained flow matching model. This can be used to incorporate boundary or initial conditions into PDE solutions generated or sampled with a generative AI model. While they briefly touch on operator learning, this is not a focus of this paper.
>
> In summary, we do not agree with the reviewer that our work is very similar to those papers.
>
> > - There is no conclusion section
>
> A short conclusion section will be added to the revised paper.
>
> ## Questions
>
> > 1. Why is the method tested on only stationary problem, e.g., scalar Darcy flow and stationary NS equations? Can it not work with a time component?
>
> The approach could also be used at least for constant boundary conditions prescribed for time-dependent problems. Some modifications might be required for boundary conditions that change over time. However, as the challenges with regard to imposing boundary conditions are already present in stationary problems whilst focussing on steady-state saves some additional notation, implementation effort and training times, we would argue that a focuss on stationary problems is a justified trade-off.
>
> We will clarify this in the discussion of limitations.
>
> > 2. This basis approach seems similar to finite elements - how does it compare/contrast?
>
> We struggled to understand what the reviewer refers to in this case? At least in our view, the approach is fundamentally different to finite elements, a mesh-based discretization methods based on approximations of infinite dimensional function spaces by finite dimensional subspaces. These finite dimensional subspaces often use a polynomial basis, and systems of equations are solved to determine coefficients of this basis representation to numerically approximate a PDE solution. Boundary conditions are typically prescribed by modifying matrix and right-hand side of this equation system. In this manuscript, we aim at improving PINOs and PINNs that learn unknown functions within a solutions structure, that by construction satisfies boundary conditions, instead of soft-constraining these in the PINO/PINN loss function. It is certainly possible that there is some deeper, underlying mathematical connection but if there is, it is seems far from obvious.
>
> > 3. What does fine-tuning mean in this context?
>
> We have tried to define this in the section comparing PINNs and PINOs (lines 100 to 117) but will overhaul this to improve clarity: fine-tuning here refers to using the network weights produced by operator training as the starting point for a second training process that aims to further improve the accuracy of the operator-generated solution for a specific parameter in a PINN-style approach.

---

> ### Comment · Reviewer_nPRe · 2025-08-01
>
> I would like to thank the authors very much for their detailed rebuttal. While I agree that the exact boundary conditions differ from those in Saad et al. and the methodology is distinct, Robin BCs are simply a linear combination of Neumann and Dirichlet boundary conditions, which are both covered in that work and could easily be extended to that case. I believe the BOON method should at least be compared to as a baseline.
>
> Also, just because the boundary conditions need to be stationary does not require the PDEs to be stationary and more realistic time-dependent cases can be tested.
>
>  In addition, while the constraint type may in different in Hansen et al. imposing conservation laws, it compares a general framework from any linear constraint using an oblique or orthogonal projection. Hence, the orthogonal projection to impose the constraint is the same methodology. In fact, Mouli et al., "Using uncertainty quantification to characterize and improve out-of-domain learning for pdes", ICML 2024 proposes the Operator-ProbConserv, which applies this oblique (more general than orthogonal) projection to impose physical constraints using Neural Operators, which I think is relevant. In addition in Cheng et al., the constraints are also imposed via orthogonal projection. So, I do not think there is much novelty in the orthogonal projection approach.
>
>  I appreciate the focus on enforcing hard constraints in this work and believe it could be beneficial to the community with a revised future submission that takes these points into consideration. At this time, I will be maintaining my score.

---

> ### Author Response · Authors · 2025-08-04
>
> > Also, just because the boundary conditions need to be stationary does not require the PDEs to be stationary and more realistic time-dependent cases can be tested.
>
> We did not mean to imply that stationary boundary conditions cannot be applied to time-dependent cases. But we would like to point out that stationary problems are relevant in many applications, not least because of the substantial cost of solving transient dynamics. We would disagree that stationary problems are less realistic. In our view it very much depends on what the relevant time scales of the modeled problem are.
>
> As the focus is on the methodology of enforcing boundary conditions we decided to stick to stationary PDEs. We want to stress, however, that this restriction does not make the problems unrealistic. Time-independent PDEs are used in applications, e.g., electrical impedance tomography is modeled by an elliptic PDE similar to the Darcy flow problem.
>
> >  I believe the BOON method should at least be compared to as a baseline.
>
> We provide the following reasoning for not comparing against BOON in this submission:
>
> 1. BOON seems tailored for mapping initial conditions to PDE solutions (p. 3 in Saad et al., 2023), and uses this in the derivation of the kernel modifications (Appendix B in Saad et al., 2023). In our examples we learn the mapping from material parameters to solutions, e.g., parameters governing diffusivity and source term in the Darcy flow problem, and, as mentioned above, focus on stationary PDEs.
> 2. The code accompanying the BOON paper (Saad et al., 2023) uses the same type of boundary condition (Dirichlet, Neumann, Periodic) on every boundary segment, whereas our approach is tailored towards mixing boundary conditions (e.g., prescribing Dirichlet and Robin BCs on adjacent/intersecting boundary segments, where the Sukumar ansatz fails, see Appendix B in our submission). It is to us not clear how the BOON method would be extended to this case, as different boundary segments would require different kernel modifications, raising the question of overall continuity of solutions.
>
> Thus, for a direct comparison, we would also have to deliver significant modifications to the BOON approach.
>
> There are further differences between the BOON approach and our method:
> - The BOON architecture (Fig. 6 in Saad et al., 2023) enforces boundary conditions in a latent space, after encoding and before a decoding layer. Thus it is unclear whether the boundary conditions are guaranteed to hold in physical space after the decoding layer (although the examples suggest it might). We enforce boundary conditions directly in physical space.
> - The BOON architecture requires the kernel corrections during every forward pass through the network, whereas our approach learns components in a solutions structure. While this increases the output dimension of the neural operator, it does not need further modifications at inference time.
>
> We would also like to stress that our main contribution is *not* the use of orthogonal projections (which is a common tool in applied mathematics), but a general method to hard-enforce boundary conditions on boundaries that are not $C^1$. Under certain conditions, the general local solutions structure can be simplified using orthogonal projections. Again, while Hansen et al. use projections to enforce conservation properties in a probabilistic manner they do not consider boundary conditions. Also the newly mentioned reference Mouli et al., "Using uncertainty quantification to characterize and improve out-of-domain learning for pdes", ICML 2024 does not consider enforcing boundary conditions.

---

> > ### Comment · Reviewer_nPRe · 2025-08-06
> >
> > I thank the authors for their response. While I agree that both stationary and transient problems are important, I think it would be best to test the method on both problem types or if it is a method just for stationary PDEs to list transient as a limitation or future work. The same Operator-ProbConserv approach used in Mouli et al. can be used to enforce linear boundary conditions as done in Cheng et al., ICLR 2025.

---

> ### Author Response · Authors · 2025-08-06
>
> Thank you for the comment. We agree that testing the approach also on time-dependent problems would be worthwhile but have to concede that this will not be possible as part of this paper due to both restrictions to available space as well as time. As mentioned in the rebuttal, we will clarify this in the discussion of limitations.
>
> > The same Operator-ProbConserv approach used in Mouli et al. can be used to enforce linear boundary conditions as done in Cheng et al., ICLR 2025.
>
> This paper deals with constraints for sampling from generative models, it does not consider neural operators. Their approach to enforcing boundary conditions in this setting is probabilistic and not a hard constraint as we consider. As mentioned in the rebuttal above, we will discuss this in the revised manuscript.

---

### Official Review · Reviewer_hALn · 2025-07-01

**Clarity:** 3
**Significance:** 1
**Originality:** 2
**Rating:** 2
**Confidence:** 5

**Summary:**

Machine learning techniques such as Physics-Informed Neural Networks (PINNs) and related operator learning methods are becoming increasingly popular for solving complex systems of PDEs. An important question is how to enforce boundary conditions. In this paper, the authors generalize a previous work to enable enforcing Neumann or Robin boundary conditions strongly (which is desirable for accuracy and training) even when the domain is not C^1. They conduct experiments showing the efficacy of this approach.

**Questions:**

None

**Ethical Concerns:**

["NO or VERY MINOR ethics concerns only"]

**Final Justification:**

The authors have made several rebuttals to my main comments. However I do not find them satisfactory. In short, the contribution in this paper is not about neural networks/operators per se, but about enforcing boundary conditions. While the research may well be sound, I do not think it is a good fit for NeurIPS.

**Limitations:**

Yes

**Quality:**

3

**Strengths And Weaknesses:**

Strengths: There has been much interest in PINNs and related techniques. It is now well-known that these techniques struggle to achieve high accuracy. Therefore, new methods that can improve accuracy are valuable.

Weaknesses:
1) The work is somewhat incremental in scope, as it is a direct generalization of Sukumar (2021). While it is relevant to relax the conditions required from C^1 to piecewise C^1, this is not, in my opinion, a big innovation.
2) The contributions of this work are only tangentially related to neural networks and deep learning. Namely, the main contribution is about how to enforce boundary conditions when solving a PDE via minimizing a loss function over a parametrized function space. This does not require the space to be a space of neural networks. Therefore, while I think this is an interesting contribution, it is mainly one to numerical PDEs and not to neural networks and deep learning. Hence I do not think it is well suited to NeurIPS. In my opinion, it would be far better suited to a journal on numerical PDEs/scientific computing, e.g. Journal of Computational Physics, SISC or JSC.

---

> ### Author Rebuttal · Authors · 2025-07-28
>
> ## Weaknesses
>
> > The work is somewhat incremental in scope, as it is a direct generalization of Sukumar (2021). While it is relevant to relax the conditions required from $C^1$ to piecewise $C^1$, this is not, in my opinion, a big innovation.
>
> It is important to note that the relaxation of this constraint greatly extends the applicability of the method. Many if not most domains in applications do *not* have globally smooth boundaries. Note that not even fairly simple geometries like squares or rectangles have $C^1$ boundaries (and that Sukumar's approach can break down already for those, see Figure 1 in the paper). The restriction is therefore significant and, we would argue, removing it is an important step towards making PINNs or NOs with hard-constrained boundary conditions much more practically useful.
>
> This may have not been expressed very clearly in the submitted manuscript. We will edit the introduction to emphasise the substantial benefit from removing this limitation.
>
> > The contributions of this work are only tangentially related to neural networks and deep learning. Namely, the main contribution is about how to enforce boundary conditions when solving a PDE via minimizing a loss function over a parametrized function space. This does not require the space to be a space of neural networks. Therefore, while I think this is an interesting contribution, it is mainly one to numerical PDEs and not to neural networks and deep learning. Hence I do not think it is well suited to NeurIPS. In my opinion, it would be far better suited to a journal on numerical PDEs/scientific computing, e.g. Journal of Computational Physics, SISC or JSC.
>
> We are not sure what exactly the reviewer means. Solving PDEs via minimization of a loss function is an approach that we, so far, have only seen in the context of machine learning based solvers for PDEs. Our approach does not really exploit parameterized function spaces but relies on using neural networks to approximate the unknown functions $\Psi$ in our solution structures. As the manuscript deals with the extension of well-established deep learning/neural network based techniques, it, in our opinion, fits to the scope of NeurIPS. While we agree with the reviewer that our paper might also fall into the scope of the suggested journals (SISC's machine learning section in particular), we would disagree that its focus is on the numerical solution of PDEs.

---

> > ### Comment · Reviewer_hALn · 2025-08-04
> > **Reply**
> >
> > I appreciate the authors’ reply. I take their first point. However, I still have the same opinion about the second. The authors’ reply in fact reinforces that opinion, as they themselves point our their approach “approach does not really exploit parameterized function spaces”. In other words, the contribution is not on the neural network aspect, but really the numerical PDE aspect, which was my argument. I will note in passing that solving PDEs via minimizing a loss function is not a new idea. For instance, the whole area of least-squares Galerkin/least-squares finite element methods has been around for several decades. One can interpret PINNs as a type of least-squares method for PDEs, where the usual FEM space is replaced by neural networks.

---

> ### Author Response · Authors · 2025-08-06
>
> > In other words, the contribution is not on the neural network aspect, but really the numerical PDE aspect, which was my argument.
>
> We are still not sure where the reviewer sees techniques from numerical methods for PDEs applied in the paper (except to generate training data for the NSE).
>
> >  I will note in passing that solving PDEs via minimizing a loss function is not a new idea. For instance, the whole area of least-squares Galerkin/least-squares finite element methods has been around for several decades
>
> Thanks for the explanation. We agree that the distance between finite and infinite dimensional function spaces in the Galerkin framework could be interpreted similarly to a loss function, although, in our opinion, this is at least uncommon terminology. The key difference is that in ML a local minimum is obtained via training where in FEM a global minimum is computed by means of projection computed analytically by calculus of variations. This similarity, however, holds for basically any neural networks and is not specific to our approach.
>
> Furthermore, this similarity also only applies to PINNs where the solution is approximated, as it is in FEM. Our approach also works for FNOs, as demonstrated, which approximates the solution operator and not the solution.
>
> We would like to emphasise that the orthogonal projections in our paper are a geometric tool and only remotely related to the orthogonal projection from the infinite dimensional function space to the finite dimensional subspace used in FEM.
>
> > One can interpret PINNs as a type of least-squares method for PDEs, where the usual FEM space is replaced by neural networks.
>
> Our approach is not restricted to PINNs. Also, while the reviewer's statement is true, it would also apply to all PINN-related techniques and not be specific to our approach.
>
> To re-iterate, our main focus is not approximating a PDE solution, as is done in LSFEM and PINNs, but the *solution operator* mapping parameter functions to the solution, as is done in neural operators. For this, the unknown functions in the derived solution structures are approximated by neural networks. As the submission deals with a technique to improve neural operators, an established neural network technique to approximate solution operators of PDEs, it should thus fit in the NeurIPS scope.

---

> > ### Comment · Reviewer_TWYM · 2025-08-07
> >
> > I'm sorry to chime in from the side, but I'd like to state that I do think this work fits well into Neurips' scope. The conference has a rich history of publishing PINN/NO/PINO papers, and I believe the subject of the submission is definitely within these fields.

---

### Official Review · Reviewer_TWYM · 2025-07-02

**Clarity:** 1
**Significance:** 3
**Originality:** 3
**Rating:** 4
**Confidence:** 4

**Summary:**

The paper tackles the problem of automatically enforcing boundary conditions in physics-informed neural operators (PINOs). The improvement over prior work is that the submitted algorithm enables the enforcement of Neumann- and Robin-conditions on piecewise smooth segments of boundaries (that are not globally smooth), such as for solving PDEs on an L-shaped domain.
The method follows the construction outlined by Sukumar et al., but separates distance functions from normalised functions and adjusts the terms in the candidate function accordingly. Numerical experiments compare the proposed method to the baseline of weakly enforcing boundary conditions on standard test problems. The results are positive, especially for the version of the PINO that uses orthogonal projections.

**Questions:**

- Line 58: If the distance function is $C^0$ or $C^1$, the Laplacian is not well-defined, or have I missed something?
- Equation (11): I would appreciate a different symbol choice for normalisation; $N(x, \phi)$ strongly resembles a Gaussian distribution. But this is subjective.

**Ethical Concerns:**

["NO or VERY MINOR ethics concerns only"]

**Final Justification:**

My original review emphasises that I like the contributions of the paper, but that clarity was a major concern. The rebuttal promises a number of changes which alleviate this issue considerably, so I increased my score.

**Limitations:**

I appreciate the open discussion of the limitations of the work in lines 85f.

**Paper Formatting Concerns:**

/

**Quality:**

2

**Strengths And Weaknesses:**

## Strengths
The paper proposes a solution to a well-defined problem that prior work suffers from: extending the method by Sukumar et al. to handle Robin and Neumann conditions on segmented boundaries. Furthermore, the article contains precise, albeit sometimes dense, instructions for implementing the proposed method, which I believe will enhance reproducibility.

That said, I think the paper's clarity could be improved considerably, which is why I ultimately recommend a (borderline) reject despite the value of the contribution. If the clarity-related issues can be resolved, I am open to revising my evaluation.

## Weaknesses

The primary concern I have is with clarity and organisation, which I find significantly limits the accessibility of the contributions. Concretely, explanations of important steps are moved to the appendices (for no apparent reason), and mathematical symbols are often defined many paragraphs after their first usage. Here are some examples, with corresponding suggestions for improvement:
- Line 67: Moving the content of Appendix A to the main paper would make the work accessible to an audience that is not necessarily familiar with the work by Sukumar et al.. For example, I found Appendix A helpful in reading the rest of this submission.
- Definition 2.1: An example would be helpful. Would listing the example from Equation (31) be appropriate here?
- Line 129: It would have helped me access the contributions if lines 92-129 were a "Background" section, and the "Methodology" section started at line 129, because everything before line 129 is existing work.
- Equations (6) & (7): Expressions like those in Equations (6) and (7) would be more understandable if all mathematical symbols were introduced before their usage. For example, $\Psi$ and $u_i$ are used in Equation (6). Still, it's only mentioned in Appendix A that $\Psi$ is learnt, and $u_i$ is not defined until Section 2.1.1 below. This lack of explanation makes it difficult to appreciate the current state of Section 2.
- Line 137: "However, it is not always possible...". This point is a critical step in the motivation for the submitted method -- it would be great if the submission could elaborate. For instance, an example would make this point more convincing.
- Line 151: It would be great if Appendix B were included in the main paper here (I suggest how to make space for it below).
- Line 167: Again, including Appendix D in the main text would clarify the construction and role of the normalisers.
- There are also some formal issues, such as the confusion between \citet and \citep, and referencing equations with "see (11)" instead of "see Equation (11)". Correcting these would improve the flow of the text.


To make space for moving Appendices A, B, and D to the main paper, the discussion of systems of PDEs in Section 2.2 could be moved to the appendix (except Theorem 2.3, which should probably be stated in the main paper). This way, the main paper explains the approach precisely for scalar systems, and the generalisation to systems of PDEs is in the appendix for further reference. However, this is just a suggestion, and I (of course) leave this decision to the authors.




## Summary

In summary, I believe the paper makes a valuable contribution: automatically enforcing Robin and Neumann boundary conditions for PDEs with smooth boundary segments. Still, in its current form, I find the presentation difficult to follow, which is why I lean towards rejection. If the presentation is improved (e.g. along the lines of the suggestions I made above), I would be willing to revise my recommendation.

---

> ### Author Rebuttal · Authors · 2025-07-30
>
> ## Weaknesses
>
> > The primary concern I have is with clarity and organisation, which I find significantly limits the accessibility of the contributions. Concretely, explanations of important steps are moved to the appendices (for no apparent reason), and mathematical symbols are often defined many paragraphs after their first usage.
>
> This is very likely due to an admittedly somewhat last-minute finalisation of the paper close to the deadline that saw some reshuffling of parts to match the page limit. Unfortunately, it seems we were not as thorough as we should have been. As part of the revision, we will do a careful proof-reading to ensure that all symbols are explained at the points where they are used for the first time.
>
> > Here are some examples, with corresponding suggestions for improvement:
>
> > - Line 67: Moving the content of Appendix A to the main paper would make the work accessible to an audience that is not necessarily familiar with the work by Sukumar et al.. For example, I found Appendix A helpful in reading the rest of this submission.
>
> We fully agree with the reviewer - the reason it was moved to the Appendix is that it is basically a repeat of a different paper and we wanted to focus on what is novel. But we agree that at least parts of Appendix A should be moved to the main paper as part of a revision.
>
> > - Definition 2.1: An example would be helpful. Would listing the example from Equation (31) be appropriate here?
>
> Yes, Equation (31) shows what is probably the most simple type of solution structure and we agree that it would serve as an instructive example for the definition. We will move the equation from the Appendix to the main paper and point it out as an example.
>
> We will also emphasise, directly after the definition, that choices other than FNO can be used to train the unknown function $\Psi$ in the solution structure, cf. the reply to reviewer [GieS](https://openreview.net/forum?id=yT8hGWqoBm&noteId=DnERyVZQ9K).
>
> > - Line 129: It would have helped me access the contributions if lines 92-129 were a "Background" section, and the "Methodology" section started at line 129, because everything before line 129 is existing work.
>
> That is a good suggestion that we'd be happy to adopt in the revised paper. Separating between "Background" (previous work") and focussing the "Methodology" on what is new in the paper will hopefully help to improve clarity of presentation.
>
> > - Equations (6) & (7): Expressions like those in Equations (6) and (7) would be more understandable if all mathematical symbols were introduced before their usage. For example, and are used in Equation (6). Still, it's only mentioned in Appendix A that is learnt, and is not defined until Section 2.1.1 below. This lack of explanation makes it difficult to appreciate the current state of Section 2.
>
> We fully agree and apologize for the oversight (see also comment above). This will be fixed in the revised paper: we will take care to introduce every mathematical symbol at the point it is first used in the paper.
>
> > - Line 137: "However, it is not always possible...". This point is a critical step in the motivation for the submitted method -- it would be great if the submission could elaborate. For instance, an example would make this point more convincing.
>
> This was indeed not very precise. We will add the following example to the Appendix to illustrate this point more clearly. Consider the L-shaped domain shown in the Appendix in Figure 6. Assume we could find a normalized distance function $\phi$ for $\Gamma_4$ that is $C^1$ everywhere, including in the point $D$.
>
> In this case, we could expand $\phi$ as
> $$\phi(D + t (-1,1)^{\rm T} = \phi(D) + t(-1,1) \nabla \phi(D) + \mathcal{o}(\left\| t(-1,1)^{\rm T} \right\|)$$
> for $t \in \mathbf{R}$. We have $\phi(D) = 0$ since it is a distance function and $\nabla \phi(D) = (0, -1)^{\rm T}$ since it is normalized with respect to $\Gamma_4$ and $D \in \Gamma_4$. Therefore, the expansion above becomes
> $$\phi(D + t(-1,1)^{\rm T}) = 0 + t(-1,1)(0,-1)^{\rm T} + \mathcal{o}(\left\| t(-1,1)^{\rm T} \right\|) = -t + \mathcal{o}(|t|).$$
> Therefore, for $t > 0$ small enough, we have
> $$\left| \phi(D + t(-1,1)^{\rm T}) + t \right| < t.$$
> However, if $t$ is chosen small enough such that $D + t(-1, 1)^{\rm T} \in \Omega$, we have $\phi(D + t(-1,1)^{\rm T}) > 0$. Therefore,
> $$\left| \phi(D + t(-1,1)^{\rm T}) + t \right| > t.$$
> This is a contradiction and hence no such function $\phi$ can exist.
>
> > - Line 151: It would be great if Appendix B were included in the main paper here (I suggest how to make space for it below).
>
> We agree (see also reply below) and will restructure the paper as suggested.
>
> > - Line 167: Again, including Appendix D in the main text would clarify the construction and role of the normalisers.
>
> See reply above.
>
> > - There are also some formal issues, such as the confusion between \citet and \citep, and referencing equations with "see (11)" instead of "see Equation (11)". Correcting these would improve the flow of the text.
>
> We only used citet as this was mentioned in the guidelines but agree with the reviewer that properly distinguishing between citet and citep will help improve the text. This will be remedied in the revision.
>
> > To make space for moving Appendices A, B, and D to the main paper, the discussion of systems of PDEs in Section 2.2 could be moved to the appendix (except Theorem 2.3, which should probably be stated in the main paper). This way, the main paper explains the approach precisely for scalar systems, and the generalisation to systems of PDEs is in the appendix for further reference. However, this is just a suggestion, and I (of course) leave this decision to the authors.
>
> We very much agree that these changes would improve vastly the clarity of the presentation. However, it would also move one of the main technical results from the paper into the Appendix. It was our belief that this is generally not desired but we would be very open to restructure the paper this way as we strongly agree that it helps to improve clarity.
>
> ## Questions
> > - Line 58: If the distance function is $C^0$ or $C^1$, the Laplacian is not well-defined, or have I missed something?
>
> This was indeed not phrased very clearly and also somewhat oversimplifies the underlying reasons why a $C^0$ distance function is problematic, which are explained in more detail by Sukumar et al.
>
> We would suggest to simply delete the sentence starting in line 58 and rephrase the statement to *"As pointed out by Sukumar, et al. (2021), suitable approximate distance functions for second-order problems should be $C^1$."* The demonstration of the problem this can cause will remain (lines 69 to 74 and Figure 1), of course, but a reader interested in the deeper reasons would have to consult the Sukumar et al. paper.
>
> > - Equation (11): I would appreciate a different symbol choice for normalisation; strongly resembles a Gaussian distribution. But this is subjective.
>
> We had not considered this, but agree. We will change the calligraphic $\mathcal{N}$ to a normal $N$ to prevent this association. However, we would suggest sticking to the letter $N$ as it helps to remember that this is the normalizer.

---

> > ### Comment · Reviewer_TWYM · 2025-08-01
> >
> > Thank you for the explanations. A lack of clarity due to last-minute reshuffling of the paper is not ideal, but I appreciate the transparency. Thank you also for replying to my suggestions, I have the impression that we agree in all cases.
> >
> > About Appendix A: I understand this sentiment, but it would be nice if the submission were understandable without assuming that every reader is familiar with all prior work. So I think mentioning Appendix A prominently would help the accessibility of the paper.
> >
> > About \citet: As far as I remember, the formatting guidelines mainly mention that \citet exists (and should be used), not that it should be used exclusively. (I think it should not be used exclusively.)
> >
> > In general, I like all promised changes. As my original review mentions, I am open to revising my score if the submission's clarity can be improved. It would have been nice to have a concrete list of changes, but I see that they might involve serious reordering of the paper, so perhaps this list wouldn't be as readable. Therefore, I will trust the authors that these changes are implemented adequately and raise my score from a 3 to a 4. After reading the other reviews, I am not sure this will be enough for having the work accepted, but I still like the contributions in this submission.

---

### Note · Authors · 2025-08-12

Two main concerns by the reviewers, accessibility and lack of a third example, can be rectified as part of the revision as outlined in our responses. Preliminary results were shown in the rebuttal. Regarding the disagreement about whether the work is within the scope of NeurIPS, we could not convince reviewer hALn that our work focusses on machine learning techniques to solve PDEs. Our assessment was, however, supported by reviewer TWYM.

While we agree with reviewer nPRe that studying time-dependent PDEs is important, we stand by our reasoning that steady-state problems are in no way inherently less realistic and that they are well suited to demonstrate efficacy of our approach.

Lastly, the papers mentioned by reviewer nPRe will be added to the related work section. However, we believe that we could convincingly explain in our responses that, upon closer reading, none of them limit the novelty of our work.

---

### Decision · Program_Chairs · 2025-09-17

**Decision:**

Reject

**Comment:**

The paper addresses the enforcement of Neumann and Robin boundary conditions in Physics-Informed Neural Operators (PINO) on domains with piecewise smooth boundaries, extending an existing framework. Two approaches are introduced: a generalized local solution structure (GLSS) that relaxes normalization requirements for distance functions, and an orthogonal projection (OP) method that can reduce the number of unknown functions in special cases. Numerical experiments on scalar Darcy flow and stationary Navier–Stokes equations indicate improved accuracy compared to weakly enforced boundary conditions.

Strengths according to the reviewers: The limitations of previous strong enforcement techniques for non-$C^1$ boundaries are well articulated. The proposed constructions are mathematically consistent, and the experimental results, though limited in scope, show promising accuracy improvements. If polished and extended, the methods could be useful in both PINN and neural operator contexts, particularly for mixed boundary conditions.

Several major open points lead me to suggest rejection of this paper:
 - Several recent works (Saad et al. 2023; Hansen et al. 2023; Cheng et al. 2025; Mouli et al. 2024) already address hard constraint enforcement—often including orthogonal/oblique projections—and were not initially cited or compared against. The rebuttal acknowledges these overlaps but does not convincingly differentiate the novelty. Similarly, the improvement upon Sukumar et al. (2021) is seen as too narrow. No comparison to some of the most relevant baselines (e.g., BOON) is provided.
 - Only two PDEs are used as examples.
- As noted by multiple reviewers, the current organization is difficult to follow, with critical definitions deferred to appendices and symbols introduced late. In particular, multiple major misunderstandings had to be clarified in the review process. While the rebuttal outlines a significant reorganization, the current submission would require major restructuring to reach acceptable clarity. This goes beyond minor revision and into the territory of a different submission, and includes additional examples.